# The Contribution of the Double Rib Contour Sign and the Rib Index to the Study of Scoliogeny, Thoracic Deformity, Progression, Outcome of Treatments and Costoplasty for Idiopathic Scoliosis

**DOI:** 10.3390/healthcare13091014

**Published:** 2025-04-28

**Authors:** Theodoros B. Grivas, Anastasios G. Christodoulou, Evangelos A. Christodoulou, Galateia Katzouraki, Marios G. Lykissas, Panayiotis J. Papagelopoulos, Elias C. Papadopoulos, Sotirios Papastefanou, Nikolaos Sekouris, Panayotis N. Soucacos, Konstantinos C. Soultanis, Elias Vasiliadis

**Affiliations:** 1Department of Orthopedics & Traumatology, “Tzaneio” General Hospital of Piraeus, 185 36 Piraeus, Greece; 2First Orthopaedic Department, Aristotle University of Thessaloniki, 541 24 Thessaloniki, Greece; an.christodoulou@gmail.com; 3St. Vinzenz Hospital, 40225 Düsseldorf, Germany; vangelchristodoulou@gmail.com; 4Spinal Department of Hygeia Hospital, 4 Erythrou Stavrou, 151 23 Maroussi, Greece; gkatzouraki@hotmail.com; 5Metropolitan Hospital, Ethnarchou Makariou 9 & El. Venizelou 1, Neo Faliro, 185 47 Pitaeus, Greece; lykissasm@gmail.com; 6First Department of Orthopedic Surgery, Athens University Medical School, “Attikon” University General Hospital, Rimini 1, 124 62 Athens, Greece; pjporthopedic@gmail.com; 7Orthopaedic Research & Education Center, “Attikon” University General Hospital, 124 62 Athens, Greece; pkalogera@gmail.com; 8Hygeia Hospital, 4 Erythrou Stavrou, 151 23 Maroussi, Greece; hpapado@yahoo.com; 9Athens Medical Centre & Mitera Children’s Hospital, Vasilissis Sofia’s Avenue, 115 28 Athens, Greece; info@totalspine.gr; 101st Department of Orthopedics, P. & A. Kyriakou Children’s Hospital, 23 Levadeias, 115 27 Athens, Greece; nick_sekouris@yahoo.com; 111st Department of Orthopaedics, School of Medicine, National & Kapodistrian University of Athens, 124 62 Athens, Greece; ksoultanis@otenet.gr; 123rd Department of Orthopaedics, School of Medicine, National and Kapodistrian University of Athens, KAT Hospital, 165 41 Athens, Greece; eliasvasiliadis@yahoo.gr

**Keywords:** idiopathic scoliosis, treatment, progression, double rib contour sign, DRCS, rib index, RI, segmental rib index, SRI, thoracic deformity, rib hump, scoliogenesis, costoplasty

## Abstract

This opinion article refers to the “double rib contour sign” and to the rib index (DRCS and RI), to their reliability study results in the chest radiographs of a control group and to their validity study results. These two parameters were introduced by the first author in this report. The introduction of the Segmental Rib Index (SRI) and its relation to spinal deformity is also discussed. The RI has been confirmed to be a strong surrogate for scoliometric readings in idiopathic scoliosis (IS). The clinical applications of the RI are analyzed for the following: (a) the documentation of deformity; (b) the assessment of physiotherapy outcomes (PSSEs); (c) the documentation of the outcomes of brace treatment; (d) the documentation of the pre- and post-operative assessment of thoracic deformity correction in different types of instrumentation; (e) its usage in prognosticating accelerated deterioration in skeletally mature adolescent idiopathic scoliosis (AIS) curves of 40–50 degrees; and (f) its usage in the recognition of the proper rib level for thoracoplasty/costoplasty. The emerging etiological–scoliogenic implications from the use of the DRCS and RI are described. The rotation of the trunk and vertebral bodies as interrelated, but distinct parameters are finally analyzed.

## 1. Introduction

The introduction of the Cobb angle in 1948 has standardized the assessment of spinal deformity in the coronal and sagittal plane in posteroanterior (PA) and lateral spinal radiographs [1]. For vertebral rotational deformity in the transverse plane, the spinous process position on the pedicle shadow was used by Nash and Moe in plane spinal PA radiographs to assess the vertebral rotation [2]. Perdriolle introduced his method to assess torsion of the vertebral body rather than rotation, using a specific template which assesses it through a line passing down the center of the convex rotated pedicle of the apical vertebral shadow [3,4,5]. Bunnell introduced the use of the scoliometer to measure trunk rotation in IS, an instrument used routinely since then in school scoliosis screening [6]. Lykyssas et al. 2015 noted, “No such methods had been previously purposed to assess rib prominence by radiographic methods, and Grivas et al. 2002, introduced the RI method extracted from the double rib contour sign (DRCS) to evaluate rib hump deformity, (RHD), in IS patients, attempting to create a safe reproducible way to assess the RHD, based on lateral radiographs” [7,8]. For historical reasons, it is reported that this method initially was presented at the 25th Anniversary Symposium on Spine Diseases at the “N. Giannestras, P. Smyrnis” meeting (Patras: Hotel Porto Rio, 1999) [9].

This report is a review article summarizing the knowledge achieved by the usage of the DRCS and the RI for the study of the thorax in normality and deformity. Additionally, it can be characterized as the authors’ opinion article, as the outcomes of using the DRCS and the rib index have introduced very useful concepts on scoliogeny in IS that will be described below.

## 2. The Presented Sign and Index (DRCS and RI)

In Scoliosis Clinics at the Thriasio General Hospital of Greece, during the clinical assessment of asymmetric children referred for IS from the school scoliosis screening (SSS) program of the hospital with the Adams test, it was noticed by TBG that the contours of the two hemithoraces were always overlapping one over the other; in other words, they were shown to be asymmetrical in the lateral spinal radiographs. This observation was systematically noted, no matter if the spines of these children were found scoliotic or not in the prescribed radiographs. This overlap was coined the “double rib contour sign” (DRCS) [9], as shown in Figure 1. Consequently, the need for quantification of the degree of this overlap, which is the asymmetry of the DRCS—in other words, the thoracic deformity in the transverse plane—triggered the introduction of the RI [9], as shown in Figure 2. The use of the index prevents metric errors due to the varying magnifications of the films depicting the thorax. This novel sign and index were presented in Greece in 1999 and in France in 2000, and published in 2002 [9,10,11]. However, this publication [11] focused merely on the implications of the DRCS in the etiology of IS.

## 3. Reliability Study for RI in Chest Radio Graph Sofa Control Group

A detailed analysis of lateral chest radiographs (LCRs) was conducted to address the following questions:Is the RI consistent across successive radiographs of the same individual?Does the same examiner obtain the same RI value in repeated measurements of the same radiograph?Does a second examiner measure the same RI value in the same radiograph of the patient?

Seventy randomly selected patients admitted to the hospital’s medical department for lung diseases (primarily pneumonia or other communicable lung conditions) between 2009 and 2013 were initially included in the study. Of these, 49 were deemed suitable for assessment. All radiographs were performed using the same technician and method. The results indicated that the RI value remained consistent across successive radiographs for each patient. Repeated measurements of the same radiograph yielded identical RI values. Additionally, the difference between the measurements taken by the two observers was statistically insignificant within the 95% confidence interval. Therefore, it can be concluded that the RI value of each patient’s lateral radiograph remained unchanged [8].

## 4. Validity Study

Another critical question was whether the distance between the radiation source and the examined child affected the RI or not. A geometrical model was used to answer this question. The dimensions of the rib cage of a growing child were used [13]. This validity study demonstrated that the DRCS is not an artifact, but substantially true. Thus, the RI is not practically affected by the distance between the radiation source and the irradiated child, provided that the radiography is always performed in any hospital or laboratory in a standardized and similar way [14,15].

## 5. Segmental RI for Spinal Deformity

Originally, the RI was assessed at a selected thoracic level where the DRCS was at its maximum asymmetry. It was noticed that in the mild and moderate IS lateral spinal standing radiographs, the level of maximum asymmetry of the DRCS varied on a vertebral level in the different types of IS. This observation triggered us to assess the RI segmentally at all thoracic vertebral levels (T1–T12) and to evaluate the association of the Cobb angles of the various IS curves with the thoracic level of the RHD. A segmental RI (SRI) study of the ribcage from T1 to T12 was then implemented and published [12], as shown in Figure 3. This study included children and adolescents who were suffering from mild to moderate IS and additional asymmetric pairs in the thorax, but not scoliotic. The measurements included scoliometric readings of the angle of trunk rotation (ATR) for truncal asymmetry (TA), the Cobb angle assessment and the SRI from T1 to T12. The SRIs of the mild and moderate thoracic, thoracolumbar and lumbar curves were then presented for the first time. An SRI value of ≥1.45–1.50 was characterized as significant, namely signifying remarkable rib asymmetry and probably prognosticating the curve progression.

In females with thoracic curves, a pattern of SRI asymmetry was present from T3 to T10 (RI = 1.59–1.75); in the thoracolumbar curves, there was a pattern from T2 to T5 (RI = 1.46–1.67), while in the lumbar curves, the RI did not exceed a value of 1.45 at all levels. In thoracic scoliotic girls, not only was the statistical significance of the correlation of the SRI with the ATR stronger, but also in more vertebral levels (T6–T12). For the thoracolumbar and lumbar curves, this correlation was not significant at any level. This implies the leading role of the rib cage, especially for the development of thoracic spinal deformity in girls.

In males with thoracic curves, the pattern of SRI asymmetry was from T6 to T11 (RI = 1.51–1.75), i.e., in lower levels compared with the female pattern of the RI in the thoracic curves. For the thoracolumbar curves, the pattern of SRI asymmetry was from T3 to T5 (1.50–1.52) and T7 to T12 (1.58–1.70), i.e., in a much lower and more extended number of rib-pair levels compared with the RI pattern of the females. This was a novel finding, indicating that the boys’ thoracolumbar curves were linked to caudal (lower thoracic levels) RC asymmetry, while in the girls’ thoracic curves, a more cephalad RC asymmetry is observed. In the boys with lumbar curves, the RI did not exceed a value of 1.44.

Additionally, the SRIs and Cobb angle correlations in the females showed significant differences between the thoracic, thoracolumbar and lumbar curve groups at the T8, T9 and T12 vertebral levels, whereas the post hoc analysis (multiple comparison tests) showed a significant difference between the thoracic and lumbar groups at t8, a significant difference between the thoracic and thoracolumbar groups at T9 and a significant difference between the thoracolumbar and lumbar groups at T12. The same analysis in the IS males only showed no significant difference between the groups. The findings of this research emphasize the importance of the role of ribcage asymmetry in relation to spinal deformity, mainly in girls [12].

## 6. The RI Is a Strong Surrogate for Scoliometric Reading in IS

A study of 66 IS subjects, with a mean age of 12.2 ± 2.9 years; 18 boys and 48 girls; and 20 thoracic, 22 thoracolumbar and 24 lumbar curves was implemented. Standing lateral spine radiographs (LSRs) were obtained, and the SRIs from T1 to T12 were assessed. The ATRs were documented using a scoliometer. In all 66 cases with IS, the scoliometer readings (ATRs) were significantly correlated to the SRI at the T6, T7 and T8 levels. In the thoracic curves, the SRI and ATR correlations were significant for the levels T6–T12. It has been observed that at the thoracic, thoracolumbar and lumbar levels in both males and females, the average trunk asymmetry (TA) decreases when shifting from a flexed to a standing position [16]. Therefore, if the TA is measured using a scoliometer during the Adams forward-bending test, the detected asymmetry would likely be greater than that obtained via the SRI method during a standing lateral surface rotation (LSR). This supports the idea that the SRI, which shows a significant correlation with scoliometric TA assessments, serves as a reasonably accurate surrogate. Hence, in the absence of scoliometric measurements, the SRI can be reliably used as an alternative for evaluating trunk asymmetry [17].

## 7. Clinical Applications of RI

The RI has many applications in the management of IS, such as a. the documentation of deformity; b. the assessment of Physiotheraputic Specific Scoliosis Exercises (PSSEs); c. the assessment of brace treatment; d. the pre- and post-operative assessment of thoracic deformity correction in different types of instrumentation; e. prognosticating the accelerated deterioration of skeletally mature adolescent IS curves of 40–50 degrees; and f. the recognition of the proper rib level for thoracoplasty/costoplasty.

### 7.1. Documentation of Deformity

Using the RI, the transverse plane of thoracic deformity from lateral spinal radiographs can be assessed and documented, avoiding any other specific radiological examinations and minimizing the hazards of radiation.

### 7.2. Assessment of Physiotheraputic Scoliosis Specific Excersises (PSSEs)

The RI, an objective tool to track RH deformity changes, was applied in the non-surgical treatment of IS using PSSEs. One study reported its use in a progressive AIS patient (Risser 4) treated with the Schroth method [18]. Initially, the RI was 1.658, with a 45° Cobb angle. As the condition worsened, they increased to 2.352 and 56°, indicating RH deformity and IS progression. After two years of daily PSSEs, the RI dropped to 1.665 and the Cobb angle to 42°. It was concluded that the RI can serve as an additional objective method to evaluate RH deformity improvement using PSSEs [18].

### 7.3. Assessment of Bracing

The rib index (RI) was used to evaluate the initial correction of rib hump deformity (RHD) in adolescents with IS treated with the Dynamic Derotation Brace (DDB). Trunk deformity (TD), including RHD, is a major concern for scoliotic children and their parents. Brace treatment targets not only spinal alignment, but also thoracic TDs like RHD. Twenty children with right thoracic (*n* = 14) and double curves (right thoracic/left lumbar, *n* = 6) meeting the SRS/SOSORT brace criteria were studied [19,20,21,22]. The reference vertebrae for RI measurement were recorded: T8 (*n* = 4), T9 (*n* = 2), T10 (*n* = 4), T11 (*n* = 6), L1 (*n* = 2) and L2 (*n* = 2). The mean thoracic Cobb angle was 27.5°. The RI decreased significantly from 1.864 before bracing to 1.205 shortly after (*p* = 0.007), indicating that the DDB effectively improved RHD in both the thoracic and lumbar components of double curves during early treatment [23,24], as shown in Figure 4.

### 7.4. Pre- and Post-Operative Assessment of Thoracic Deformity Correction in Different Types of Instrumentation

Using the RI in AIS, the impact on thoracic deformity pre- and post-operatively was assessed, as well as the RCD deformity changes—correction on the transverse plane with or without costoplasty, using various surgical techniques and types of instrumentation [7,25,26,27,28,29,30,31,32]; further assessments investigated how the implant density impacted three-dimensional deformity correction in AIS with the Lenke 1 and 2 curves, treated by posterior spinal fusion without Ponte osteotomies [33], and how the study of the relationships between the different torsion-related parameters measured on 2D radiographs can indirectly guide the clinician, concerning the torsion of a given curve [34].

### 7.5. Prognosticating Accelerated Deterioration in Skeletally Mature Adolescent IS Curves of 40–50 Degrees

The RI method was used to investigate the prognostic significance of simple radiographic rotational parameters, and to identify the AIS curves of 40–50 degree Cobb angles with accelerated deterioration following skeletal maturity. A threshold value of 1.915 for the RI at maturity was achieved, discriminating fast progressors. Thus, the RI may be useful to predict fast progression in 40–50° AIS curves following skeletal maturity, with the curves indicated for early fusion [35].

### 7.6. Recognition of Proper Rib Level for Thoracoplasty/Costoplasty

In publications reporting the outcomes of spinal operations for IS, it is interesting to note that there is always remnant/residual RHD [36]. The scoliogenic importance of this phenomenon will be analyzed in the next paragraphs. The only effective way to correct residual rib hump deformity (RHD) is costoplasty, which is rarely performed due to various reasons. In cases of idiopathic scoliosis requiring surgery and presenting with significant RHD, thoracoplasty is sometimes added for spinal correction. However, the outcomes are often unsatisfactory due to persistent RHD, leaving patients and families disappointed [37]. Erkula et al. (2003) [38] attributed this to the difficulty in identifying the precise vertebral level of the rib hump, as the ribs slope downward obliquely. In severe RHD, determining the vertebra corresponding to the highest point of the hump is challenging. They recommended using a scanogram or 3D reconstruction of the spine and ribs to accurately locate the hump’s corresponding vertebral level [38]. The usage of the SRI method can serve successfully to sort out this costoplasty problem [12].

## 8. Etiological—Scoliogenic Implications Using DRCS and RI: Rotations of Trunk and Vertebral Bodies Are Interrelated, but Distinct Parameters

Lertudomphonwanit et al. 2021 reported that there is limited evidence on vertebral rotation and RHD regarding pedicle screw density [33]. Pedicle screw constructs have been shown to improve vertebral rotation correction and lessen RHD compared with hook–rod instrumentation [39], yet the question raised is whether truncal rotation (ATR) is a result of the vertebral rotation only, or if there is an additional component contributing to the truncal rotation. The answer to this question has been provided in a number of peer-reviewed publications.

Implementing their school scoliosis screening programs, both Nissinen et al. 1993 [40], using the scoliometer, and Willner 1983, 1984, 1989 [41,42,43], using the Moiré method, found a number of children with truncal asymmetry (hump), but without any curve in spinal radiographs. Willner 1984 [41] noted that “in former Malmo studies these small asymmetries of the trunk were not related to a lateral deviation of the spine, seen roentgenographically, exceeding 9 degrees”. Nissinen et al. 1993 [44] stated that “The predictive significance of baseline trunk asymmetry was independent of all the other determinants entered in the multifactorial logistic model. The relative risk (odds ratio) for an increase of 1 mm of hump size was 1.72 (9 5% confidence interval [Cl]—1.39–2.12) in boys and 1.55 (95% Cl = 1.33–1.82) in girls. Thus boys with humps of 6 mm had approximately a fivefold risk of developing scoliosis as compared with boys having a symmetric trunk (hump = 0 mm) at the age of 10.8 years”. It was also stated that “hump size was found to be the most powerful predictor of IS development” [44].

In our scoliosis clinic, when examining the referred children from our screening program, about 30% of the younger girls referred (aged less than 13 yrs. old with an ATR ≥ 7°) were found to have either a straight spine or a spinal curve with a Cobb angle less than 10°. The possible cause of existence of the phenomenon of RHD without spinal deformity was assessed in a later study on referrals from our school screening program, correlating the RI with the Cobb angle [45]. In younger individuals, the correlation between clinical deformity (trunk/thoracic asymmetry via the RI) and radiographic findings (the Cobb angle) was not statistically significant, unlike in the older screened girls (14–18 years), where it was. This highlights growth as a key factor in the link between the thoracic and spinal deformities in girls with IS [45]. The results also suggest that rib hump deformity (RHD) precedes spinal curvature in the development of mild to moderate IS. Therefore, younger individuals with thoracic surface deformity but no spinal curve should be closely monitored during growth and not discharged from follow-up.

The study of the outcomes of the surgical treatment of IS provides additional strong evidence that thoracic deformities in terms of the thoracic (truncal) rotation and the vertebral rotation are interrelated, but distinct parameters [36]. Post-operatively, the hump is corrected incompletely, and it recurs or worsens during follow-up and even more intensively in skeletally immature operated scoliosis children. The key reason for this phenomenon is the fact that RH deformity (RHD) is mainly due to the asymmetric development of the ribs, and much less so due to the rotation of the vertebrae in the thoracic spine. Surgery on the spine cannot limit the asymmetry of the ribs or stop the mechanism that causes the asymmetrical growth, which is probably the result of an asymmetric autonomous nervous system (ANS). Moreover, the existence of a residual rib hump, reported in a review of articles describing the surgical outcomes for IS, supports the important protagonistic role of RHD on scoliogenesis, which precedes the subsequently formed spinal deformity [36]. Our above-described concept is in line with that reported by Prof. Sevastik’s research work, pertinent to scoliogeny, emphasizing the important role of the rib cage in scoliosis development. For that reason, the usage of the RI in our referred studies shed more light on the theory of the possible asymmetric functioning of the ANS, reported by Prof. Sevastik and his team [46,47,48,49,50,51,52,53,54,55,56,57,58]. They also support a physiological approach to the surgical treatment of progressive early IS, which was proposed by Sevastic [59].

In the literature, 3D analysis is increasingly used to study scoliotic curve morphology [60,61,62,63,64]. However, the RI and DRCS rely on 2D radiographs, which have inherent limitations when based solely on the coronal or sagittal planes. Despite this, key radiological parameters—such as Cobb angles, Mehta RVAs and Perdriolle angles—are still primarily measured on standard posteroanterior (PA) and lateral radiographs. Readily available chest radiographs in medical archives, especially for children and adolescents, can be effectively used for RI and SRI assessments, minimizing radiation exposure. Additionally, the RI and SRI methods are suitable for both prospective and retrospective studies of IS treatment, provided the standard 2D imaging protocols are followed.

Currently, studies of longitudinal developed models for the prediction of the progression of IS curves also use parameters taken from 2D radiography because they need existing initial data from IS assessments, which were previously based on 2D and not 3D methods in the majority of centers [65]. Finally, transdisciplinary studies using machine learning on clinical data to develop in-house programs for predicting curve progression involve specialized terminology that may be not only challenging to digest, but also difficult to assess [66,67,68].

In conclusion, this report confirms the beneficial contribution of the use of the DRCS, the RI and the SRI to the assessment of thoracic deformity and spinal deformity and the investigation of scoliogenesis in IS.

## Figures and Tables

**Figure 1 healthcare-13-01014-f001:**
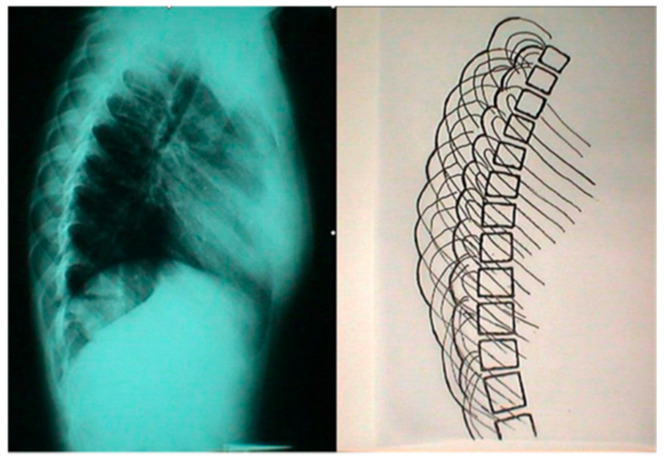
The double rib contour sign (DRCS) of the rib cage in a standing spinal radiograph (from our citation no. [12]).

**Figure 2 healthcare-13-01014-f002:**
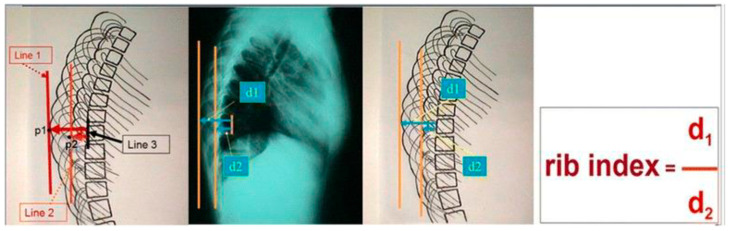
The rib index (RI) of the thoracic cage in a standing spinal radiograph (from citation no. [12]).

**Figure 3 healthcare-13-01014-f003:**
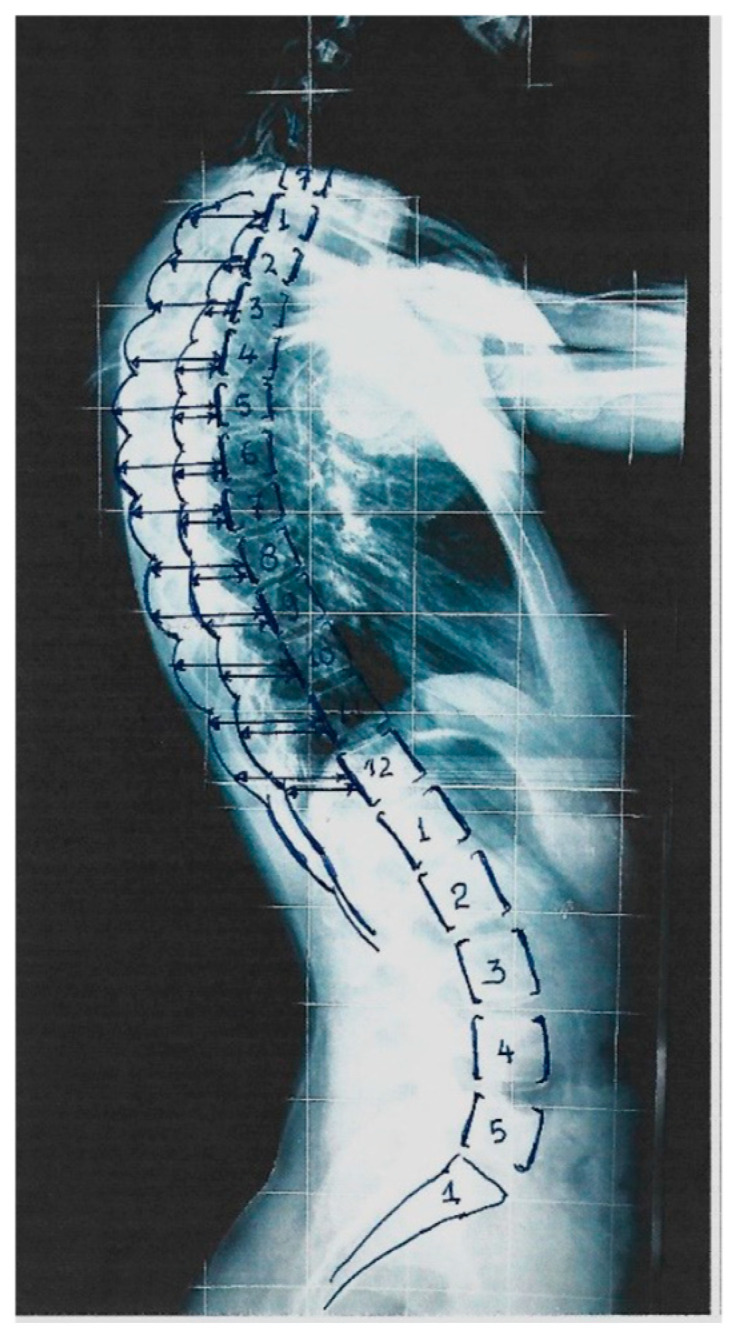
The Segmental Rib Index (SRI) of the thoracic cage in a standing spinal radiograph (from our citation no. [16]).

**Figure 4 healthcare-13-01014-f004:**
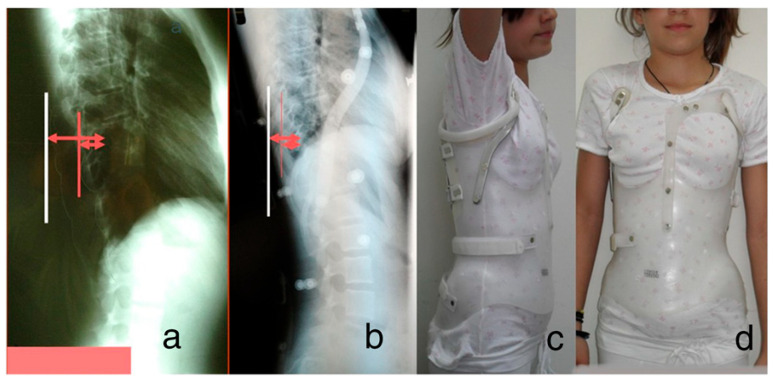
The assessment of pre-brace treatment RI and early post-brace RI. (**a**,**b**) depict the radiographs and (**c**,**d**) the clinical pictures of a girl wearing the brace (modified from our reference no. [16]).

## Data Availability

Not applicable.

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
