# Peer review of "The Contribution of the Double Rib Contour Sign and the Rib Index to the Study of Scoliogeny, Thoracic Deformity, Progression, Outcome of Treatments and Costoplasty for Idiopathic Scoliosis"

_healthcare, 2025, doi:10.3390/healthcare13091014_

Round 1

Reviewer 1 Report

Comments and Suggestions for Authors

This study explores the clinical application and etiological significance of the Double Rib Contour Sign (DRCS) and Rib Index (RI) in idiopathic scoliosis (IS), covering reliability analysis, validity verification, the proposal of Segmented Rib Index (SRI), and the value of RI in brace treatment, surgical evaluation, and prognosis prediction. While the research holds clinical relevance, further improvements are needed in methodological rigor, data presentation, and logical coherence. The following suggestions are provided for the authors:

1.Statistical Methods: Details on statistical approaches are missing. Clarify which specific methods were used for different analyses.

2.Sample Justification: Provide the rationale for the sample size (49 patients) and the criteria for participant selection.

3.Data Visualization: Present key results using charts or tables to enhance clarity.

4.Lenke Classification Correlation: Further investigate the relationship between RI values and Lenke subtypes to strengthen clinical applicability.

Comments on the Quality of English Language

The English language quality of this article is acceptable and meets the writing standards for journal articles.

Author Response

Dear chief editor thank you very much for sending me the reviewers comments

We provide analytically our answers below

Reviewer 1

This study explores the clinical application and etiological significance of the Double Rib Contour Sign (DRCS) and Rib Index (RI) in idiopathic scoliosis (IS), covering reliability analysis, validity verification, the proposal of Segmented Rib Index (SRI), and the value of RI in brace treatment, surgical evaluation, and prognosis prediction. While the research holds clinical relevance, further improvements are needed in methodological rigor, data presentation, and logical coherence. The following suggestions are provided for the authors:

1.Statistical Methods: Details on statistical approaches are missing. Clarify which specific methods were used for different analyses.

2.Sample Justification: Provide the rationale for the sample size (49 patients) and the criteria for participant selection.

3.Data Visualization: Present key results using charts or tables to enhance clarity.

4.Lenke Classification Correlation: Further investigate the relationship between RI values and Lenke subtypes to strengthen clinical applicability.

Our answer:

Dear reviewer thank you very much for your time spent to review our opinion paper

In response to your reasonable queries, all the requested details regarding statistical methods, sample sizes, tables and charts, and the relationship between rib index values and Lenke subtypes involving the thorax are thoroughly discussed in our previously published papers, which are cited in this opinion paper. Almost all of these papers are open-access and published in peer-reviewed journals, making them readily available to any interested reader. For this reason, we did not include this information in the opinion paper.

Additionally, I would like to emphasize that “opinion papers” submitted to MDPI are expected to be concise. In adherence to this guideline, we chose not to include the aforementioned details to keep the paper brief. We apologize if this is perceived as a limitation, but in reality, it is not, as all the relevant information is freely accessible for those who wish to explore the topic further.

Reviewer 2 Report

Comments and Suggestions for Authors

Dear Authors,

this is a very informative narrative review/opinion paper of renowned specialists, addressing their description, and discussion, of their own method of the assessment of spinal deformities. 

For that reason, I do not think that the role of a peer reviewer is to appraise the presented method critically but rather to study the report and provide some suggestions for its improvement.

I would ask the Authors for clarifications of their statement provided in lines 256-262, on Adams FBT test (or moire topography) and its confrontation with radiographic examination. Please clarify/elaborate whether the SSS tests have predictive values for early detection, or are indecisive, and whether an existing trunk deformation (hump) is not a surrogate for 'spinal scoliosis'. Does 'spinal scoliosis' mean 'structural deformation'? And please clarify the statement in lines 258-9 - it is unclear whether it is a citation, and from which of the papers [43-6].

ll. 310-316 - to me, the paragraph/statement is unclear: what longitudinal models do you mean, and how is it related to AI/machine learning usage? Please clarify the statement.

Minor tips:

  • l. 67, l.85 - a word seems missing in '...at 1999' - conference? please rewrite/amend
  • l. 70 - 'authors' - as there are multiple authors of this report
  • subtitles - spaces are missing (but the content is understandable)
  • l. 231 - 'peer reviewed'

Please consider the statements 'thoracic deformity' and 'spinal deformity' - to me, it could be vague for readers whether 'thoracic deformity' is to the deformity of the thorax rather that thoracic scoliosis.

Comments on the Quality of English Language

There are single mistakes which I have indicated in my report to the Authors. Just a polishing is needed.

Author Response

Dear chief editor thank you very much for sending me the reviewers comments

We provide analytically our answers below

Reviewer 2

this is a very informative narrative review/opinion paper of renowned specialists, addressing their description, and discussion, of their own method of the assessment of spinal deformities. 

For that reason, I do not think that the role of a peer reviewer is to appraise the presented method critically but rather to study the report and provide some suggestions for its improvement.

Our answer:

Dear reviewer thank you very much for your time spent to review our opinion paper and for your kind words. Also, thank you for providing some suggestions for its improvement.

-----------------------------------------------------------------------------------------------------------------------------------------

I would ask the Authors for clarifications of their statement provided in lines 256-262, on Adams FBT test (or moire topography) and its confrontation with radiographic examination. Please clarify/elaborate whether the SSS tests have predictive values for early detection, or are indecisive, and whether an existing trunk deformation (hump) is not a surrogate for 'spinal scoliosis'. Does 'spinal scoliosis' mean 'structural deformation'? And please clarify the statement in lines 258-9 - it is unclear whether it is a citation, and from which of the papers [43-6].

Our answer: Dear review thank you for asking for clarifications in lines 256-262.

This paragraph now is revised as requeste:

Implementing their school scoliosis screening programs both, Nissinen et al 1993 [43] using scoliometer and Willner 1983, 1984 and 1989 [44-46] using the Moiré method, they found a number of children with truncal asymmetry (hump), but without any curve in spinal radiographs. Willner 1984 [44] noted that “in former Malmo studies these small asymmetries of the trunk were not related to a lateral deviation of the spine, seen roentgenographically, exceeding 9 degrees”. Nissinen et all 1993[47], stated that “The predictive significance of baseline trunk asymmetry was independent of all the other determinants entered in the multifactorial logistic model. The relative risk (odds ratio) for an increase of 1 mm of hump size was 1. 72 (9 5% confidence interval [Cl] - 1.39-2.12) in boys and 1.55 (95% Cl= 1.33-1.82) in girls.  Thus boys with humps of 6 mm had approximately a fivefold risk of developing scoliosis as compared with boys having a symmetric trunk (hump = 0 mm) at the age of 10.8 years. It was also stated that “hump size was found to be the most powerful predictor of IS development [47].

--------------------------------------------------------------------------------------------------------------------------------------

  1. 310-316 - to me, the paragraph/statement is unclear: what longitudinal models do you mean, and how is it related to AI/machine learning usage? Please clarify the statement.

Dear review thank you for asking for clarifications in lines 310-316.

The paragraph was revised to make sense and match with the following lines of the text.

Currently 3D analysis is more frequently used as a procedure to study the morphology of scoliotic curves, [63-67]. The RI index and the DRCS results are based on a 2D radiography. Yet any study based exclusively on coronal or sagittal plane has its limitations.

Our paragraph stating, “Currently, studies on longitudinally developed models for predicting the progression of IS curves also use parameters derived from 2D radiography, …” was included to highlight that currently even the most advanced models for studying IS still rely on 2D parameters. This reinforces the point that the so-called "limitation" of the RI due to the use of plane radiographs does not diminish the method’s value.

Minor tips:

  • l. 67, l.85 - a word seems missing in '...at 1999' - conference? please rewrite/amend

It was amended “For historical reasons it is reported that this method initially was presented at the 25th Anniversary Symposium on Spine Diseases of "N. Giannestras, P. Smyrnis" meeting. Patras: Hotel Porto Rio; 1999 [9].”

  • l. 70 - 'authors' - as there are multiple authors of this report,

amended” as authors’ opinion article”

  • subtitles - spaces are missing (but the content is understandable)

the lay out of the page is arranged bt the MDPI

  • l. 231 - 'peer reviewed' corrected to - 'peer reviewed'

Please consider the statements 'thoracic deformity' and 'spinal deformity' - to me, it could be vague for readers whether 'thoracic deformity' is to the deformity of the thorax rather that thoracic scoliosis.

Dear respected reviewer our answer to this statement is:

Idiopathic scoliosis (IS) is both spinal and thoracic deformity. Thoracic scoliosis is the one where there are both spinal and thoracic deformity. We say so because the rib hump is actually thoracic deformity, however there are younger children with a hump – (thoracic deformity - mainly in transverse plane) but without spinal deformity that is deformity of the spine, the “central axis”. Later some of them will develop IS. A lot of authors when they write about scoliosis they consider as IS only the spine which is inaccurate. In IS both spine and thorax are deformed.

Reviewer 3 Report

Comments and Suggestions for Authors
  • This is an opinion article without main research question
  • Topic is original or relevant to the field.  There is no such summary of rib parameters in scoliosis, so I think this paper addresses this problem quite well. This is a new attitude to scoliosis pathogenesis - may be further research prove that everything starts with rib asymmetry?
  • Conclusions summarize the manuscript.
  • References are appropriate.

Rib hump and rib index (RI) are usually underestimated, so this paper is very important. It is an opinion article, quite well written, however there are some errors.

  1. line 76 - what does TBG mean?
  2. line 94 and 157 ????
  3. line 174 - what does PSSE mean ?
  4. line 230 & 236  - rather thoracoplasty not THORACOLPASTY
  5. figures should be new, not from previous published paper. 

Author Response

Dear chief editor thank you very much for sending me the reviewers comments

We provide analytically our answers below

Reviewer 3

Our answer:

Dear reviewer thank you very much for your time spent to review our opinion paper and foryour suggestions.

  • This is an opinion article without main research question Thank you
  • Topic is original or relevant to the field.  There is no such summary of rib parameters in scoliosis, so I think this paper addresses this problem quite well. This is a new attitude to scoliosis pathogenesis – Thank you for your comments.
  • may be further research prove that everything starts with rib asymmetry? further research prove that everything starts with rib asymmetry would be very valuable.
  • Conclusions summarize the manuscript. Thank you
  • References are appropriate. Thank you

Rib hump and rib index (RI) are usually underestimated, so this paper is very important. It is an opinion article, quite well written, Thank you

however, there are some errors.

Dear Reviewer

Thank you for highlighted the errors . We corrected all of them.

  1. line 76 - what does TBG mean?  (TBG = Theodoros B GRIVAS)
  2. line 94 and 157 ???? amended SRI Segmental RI
  3. line 174 - what does PSSE mean? Physiotheraputic Specific Scoliosis Exercises, was added to Abbreviations
  4. line 230 & 236  - rather thoracoplasty not THORACOLPASTY, yes corrected
  5. figures should be new, not from previous published paper. 

The figures were used to help readership understand better the methods of RI and SRI. For all of them we have the copyright.

Round 2

Reviewer 2 Report

Comments and Suggestions for Authors I have inspected both documents - the revised paper and the similarity report. I support my previous opinion regarding this review. Regarding the amendments, I think the Authors have made a good progress with the paper, in terms of the plagiarism issues:
  • The similarities (same phrases or words) are:
    • With references or
    • standard terms
    • some self-citations, which can be regarded as the Authors' style of writing,
    • technical (such as Authors' contributions, funding et.)
  • the figures are the same as in other publications from the Authors, but referencing/source is provided with each caption/legend).
   The content of the paper has many repetitions from previous publications, and the novelty of the paper is an issue. But to me, the explanations of the Authors, given the type of the paper, can be assessed as satisfactory. I am not an expert as regards the formal / legal regulations regarding plagiarism.   I hope my input will help with your decisions.

Reviewer 3 Report

Comments and Suggestions for Authors

Now it looks much better, and in my opinion it can be published.